# WEDM-Generated Slot Width Variation Modeling

**DOI:** 10.3390/mi13081231

**Published:** 2022-07-31

**Authors:** Oana Dodun, Laurențiu Slătineanu, Gheorghe Nagîț, Adelina Hrițuc, Andrei Marius Mihalache, Irina Beșliu-Băncescu

**Affiliations:** 1Department of Machine Manufacturing Technology, Gheorghe Asachi Technical University of Iași, 700050 Iasi, Romania; oanadodun@tcm.tuiasi.ro (O.D.); slati@tcm.tuiasi.ro (L.S.); nagit@tcm.tuiasi.ro (G.N.); andrei.mihalache@tuiasi.ro (A.M.M.); 2Department of Mechanics and Technologies, Ștefan cel Mare University of Suceava, 720229 Suceava, Romania; irina.besliu@yahoo.com

**Keywords:** wire electrical discharge machining, slot width, influence factors, empirical mathematical models

## Abstract

Wire electrical discharge machining (WEDM) is a process that is used when it is necessary to manufacture small-width slots with a micrometer accuracy or to precisely detach parts with complex contours from metal workpieces in the form of sheets or plates. The fact that the wire electrode rests only in the working zone in two of its guides allows it to achieve micrometric oscillations, leading to the generation of an error from the flat shape of the slot surfaces that gradually develops into the workpiece. The slot widths are influenced by several factors, such as the workpiece thickness, pulse-on time, pulse-off time, the wire tension force, the current, and the wire movement speed along its axis. Some theoretical assumptions about the behavior of the wire electrode were first considered. An experimental research plan was then designed to obtain additional information on the influence of the mentioned factors on the slot width in different positions of the cross-section through the slot. The statistical processing of the experimental results led to the elaboration of empirical mathematical models that highlight the order of influence and the intensity of the influence exerted by the factors mentioned above.

## 1. Background

Electrical discharge machining is a processing method based on the effects of electric discharges generated between the asperity peaks on the surface of the tool electrode and the workpiece surface. The active zones of the two electrodes involved in the processing are immersed in a dielectric liquid. Due to the electric discharges, small amounts of the tool electrode and the workpiece material are melted and vaporized. The micro-explosions generated by the electrical discharges throw small amounts of material detached from the electrodes into the working gap. Electrical discharges are carried out on paths characterized by a minimum electrical resistance. The recirculation of the dielectric liquid contributes to the removal from the working gap of the material particles initially detached from the two electrodes by melting and vaporization. In the following sequence, due to a working movement that reduces the distance between the electrodes, the electric discharges will affect other asperity peaks of the electrode surfaces. In this way, the removal of the material from the workpiece continues until the surfaces proposed to be obtained by such a processing method are fully formed.

A group of electrical discharge machining methods is based on using a wire-type tool electrode, which is in a continuous motion from a coil to a coil on which the wire is wound after passing through the working area. In the working area, the wire has a rectilinear shape, being adequately guided, and subjected to the action of a tensile force. Due to its rectilinear shape, wire electrical discharge machining (WEDM) allows only ruler surfaces to be obtained (flat, cylindrical, conical surfaces, in the shape of a single-sheet hyperboloid, etc.).

WEDM can only be applied to workpieces made of electrically conductive materials. Like other procedures of processing by electric discharges, an intersection with sharp angles of the surfaces of the walls to be obtained is not always an impediment to the successful application of this processing method [1,2,3,4].

In recent decades, WEDM has become a process capable of ensuring high machining accuracy. WEDM-obtained surfaces usually do not require other further processing operations, thus significantly reducing energy consumption and the duration of obtaining certain surfaces. For this reason, WEDM is valued as an environmentally friendly processing technology and is applied where appropriate in the manufacturing industry.

The above issues have led to an intensive investigation into the possibilities of expanding the use of WEDM and increasing its processing performance. The aim was to increase the removal rate of the workpiece material but also to reduce the surface roughness resulting from the machining process, and improve the machining accuracy, respectively.

Thus, an analysis of how the vibration phenomenon of the wire electrode takes place was performed by Dauw et al., the results of the research being published in 1989 [5]. According to one of the research conclusions, the inaccuracy of the WEDM process is influenced by the physical–mechanical interactions during the electrical discharge machining process.

In a paper published in 1997, Hsue et al. proposed the introduction of the concept of discharge-angle, using a mathematical expression obtained from geometric arguments [6]. They still considered the deflection of the wire electrode when developing a model to estimate the material removal rate in the case of WEDM.

Puri and Bhattacharyya developed a vibrational behavior analysis of the wire electrode [2]. They aimed to highlight the influence that the frequency of the pulses, the wire tensions, and the ratio of the so-called job height and the span of the wire between the guides exert on the maximum amplitude of the wire tool electrode vibration.

Mahapatra and Patnaik addressed the issue of optimizing the WEDM process using the Taguchi method [7]. They considered the discharge current, pulse duration, pulse frequency, wire speed, wire voltage, and dielectric flow as input factors. The objective was to identify combinations of the values of the input factors to obtain a better metal removal rate, surface finish, and cutting width.

The influence of the different forces acting on the wire electrode, which generates a deformation of it and, as such, determines the appearance of deviations from wall-flatness, was highlighted in three works whose first author was Sanchez [8,9,10]. In these works, solutions are offered to contribute to the improvement of accuracy of the WEDM process.

In a paper published in 2010, Liang et al. addressed the issue of an error in making the intersections between flat surfaces when using WEDM due to a delay in the appearance of electric discharges in the middle area of the thickness of the workpiece [11]. A digital camera was used to assess the size of the time difference in which the discharges occur in the upper and lower surface of the workpiece, respectively. The results obtained allowed the estimation of the value of the wire deflection and the formulation, as such, of an assessment of the accuracy of the machined part.

In 2013, Feng defended a doctoral thesis [1] that addressed issues relating to modeling and simulation of crater formation during micro WEDM. In this thesis and a paper previously published by Feng et al. [12], the explanation of the occurrence of deviations from the flat shape of the surface generated by WEDM was based on the vibrational phenomena capable of altering the rectilinear shape of the filiform electrode in the working gap.

An analysis of how to generate errors in the case of WEDM rough corner-cutting was performed by Chen et al. [13]. They proposed modeling the wire electrode center’s trajectory to assess the influence of some factors on the corner error.

A study of wire-tool vibration during the WEDM from the space temperature field and the electromagnetic field was developed by Chen et al. [14]. One of the objectives of the analysis was to identify solutions to reduce vibration amplitude and thus improve machining accuracy.

Dongre et al. proposed a model for optimizing the use of WEDM for silicone ingot slicing [15]. They appreciated that it is possible to minimize the vibration amplitude capable of generating deviations from the flat shape of the machined surfaces. One of the goals of the optimization was the kerf width.

In 2016, Conde et al. showed that an elliptical shape could be considered to model a winding of the vibrational movements of the wire [16]. The main half-axis would correspond to the wire delay. In contrast, the small half-axis defines the components capable of generating the concavity effect of the surface generated by processing. A model on the vibration mode of the wire electrode was proposed in 2016 by Habib and Okada [17]. They found that the concavity of the surface is greater when the radius of the path traveled by the wire electrode is smaller.

Ablyaz and Lesnikov [18] performed WEDM-generated slot width measurements. One of their experimental research objectives was to highlight the influence exerted by pulse power on the width of the slot at the level of the outer surface of the test sample.

Straka et al. aimed to investigate the geometric accuracy of the machined surface by taking into account the flatness deviations of surfaces obtained by WEDM [19]. Experimental research identified significant differences in maximum flatness deviations when using WEDM roughening, semi-finishing, and finishing.

Subsequently, in 2018, Conde et al. [20] considered that the delay effect of the wire electrode might affect the processing accuracy of WEDM. They proposed a method that could be used to evaluate such an effect quantitatively. The experimental research considered the variation of the dielectric fluid pressure, the radius of the part, the thickness of the workpiece, and the properties of the electric discharges. They concluded that the concavity of the side walls also depends on the values of the input factors of the WEDM process mentioned above.

The greater influence of wire tool electrode deformation on the accuracy of the WEDM process was also highlighted in [21] when the Elman-based Layer Recurrent Neural Network method was applied to predict the surface accuracy obtained by WEDM. The authors of the cited paper showed that when the wire diameter is small, and the thickness of the workpiece is large, the wire stiffness is reduced, and the wire deformation increases, causing a decrease in the accuracy of the processed surfaces. The coefficient of variation, determined as the ratio between the standard deviation and the average data value in a given plan, was calculated to establish the optimized machining strategy regarding processing accuracy.

Chen et al. [22] explained the generation of workpiece corner error during WEDM by considering string vibration theory. They considered it possible to reduce the workpiece corner error by using a novel surface microstructure wire electrode. Experimental research showed a possibility of reducing the workpiece corner error by 5.23–27.15% when using this electrode compared to the brass wire electrode.

Experimental research has confirmed a certain validity of the results obtained theoretically. Abyar et al. showed that electrical discharge frequency and wire tensions are input factors capable of influencing estimated wire deviation and total machining errors [23]. They found experimentally that wire deflection error can account for up to 57% of total machining error. They appreciated that the proposed theoretical models allow the prediction of wire deflection errors in arced corner machining and the use of solutions to reduce them. In another paper, the same authors proposed a calculation method corresponding to the total theoretical arced machining errors by considering wire deflection and spark gap size variation around the wire interface [24].

The deformation under the action of electromagnetic forces of the corner areas of the parts obtained by WEDM was the objective of research undertaken by Chen et al. [25]. They used finite element modeling to analyze the magnetic field distribution in the processing area. Two empirical mathematical models were proposed to highlight the influence of input factors in the WEDM process on corner deformation.

In a paper published in 2019, Zhang et al. took into account energy consumption and processing precision when evaluating the ecological character of processing by electrical erosion using wire electrodes. The results of assisting the WEDM process were investigated by a magnetic field, and they observed that it is thus possible to reduce energy consumption and thermal deformation [26]. They considered that a certain intensity of the magnetic field could ensure optimal results of the application of WEDM assisted by a magnetic field.

Sabyrov et al highlighted the possibilities of using the ultrasonic vibration-assisted WEDM process to cut thin slots [27]. They considered that the ultrasonic activation of the processing area in the case of WEDM can be beneficial from the point of view of the material removal rate and that it can lead to the appearance of elliptical shape craters.

A mechanistic model was proposed by Parthiban et al. to highlight the wire deflection during the WEDM process [28]. An experimental study was performed in which the current intensity, spark gap, wire feed, and workpiece feed speed were considered input factors. The experimental results were compared with those obtained using the proposed mathematical model.

A magnetic-assisted WEDM approach combined with optimization of processing parameters can improve the processing accuracy of WEDM-generated surfaces by decreasing the deformation in the corner area of thin-walled sharp-corner parts, according to the results of research conducted by Yan et al. [29].

Rohilla et al. conducted experimental research to investigate the influence of aluminum metal powders in dielectric liquid on the morphology of surfaces made using WEDM in the case of test parts from Monel 400 [30]. They found that it is possible to reduce the thickness of the melt layer up to 43% compared with the recast layer’s thickness generated at low and high values of the discharge parameters.

A way to optimize the WEDM process in the case of nitinol test samples by considering three input factors (pulse-on time, pulse-off time, and current) was proposed by Chaudhary et al. The material removal, surface roughness, and surface layer microhardness were used as process output parameters. Optimization was performed through a heat-transfer search algorithm [31].

The results of experimental research on the possibilities of improving the WEDM process are presented in [32,33].

The aspects mentioned above determined an intense investigation to extend the use of WEDM and increase its machining performances.

However, it was found that there is relatively little information on the variation of the slot width generated by WEDM. As such, research was developed on theoretically generating the variation of the slot width in a slot cross-section.

An experimental research program was thus designed and conducted to highlight the way of action and intensity of the influence exerted by the input factors in the WEDM process on the width of the slot measured in the area where the wire electrode enters the test sample in the middle area of the test sample thickness, and in the area where the wire electrode exits the test sample.

## 2. Materials and Methods

### 2.1. Theoretical Considerations

In a simplified version (Figure 1), the wire electrode can be considered a thin elastic element moving continuously, at a constant speed, between the lower and the upper guide. The very low stiffness of the wire both along a direction perpendicular to the desired rectilinear direction of the axial movement of the wire and its twisting when passing on different guide rollers may lead to deviations from the theoretical considerations used in this article.

In the area where the wire passes through the test sample, under the action of the working feed rate along the contour to be obtained by processing and therefore as a result of reducing the working gap to values less than the penetration distance between the test sample and the wire electrode, electric discharges appear.

The breakthrough distance *s* of the dielectric liquid is the distance between the wire electrode and test sample for which electric discharges occur along paths characterized by a minimum electrical resistance [34]:(1)$$s<\frac{U}{E},$$
where *s* is the size of the lateral gap (mm), *U* is the initial voltage of the electric discharges between the electrodes (V), and *E* is the dielectric strength of the fluid in the gap (V/m).

Typically, the value of the side gap used in electrical discharge machining is between 0.01 mm and 0.5 mm.

Suppose the wire electrode is considered to have an approximately cylindrical shape near the area passing through the plate-type test sample of thickness H (Figure 2a). In that case, the discharges will occur only on the surface of the wire electrode with which the wire electrode enters the test sample. For this surface, due to the feed rate, the distances between the wire electrode and the blank become smaller than the breakthrough distance s, a length corresponding to the thickness *H* of the blank.

Suppose an approximately uniform development of the electric discharges on that half of the cylindrical surface of the wire electrode inside the slot develops into a test sample (Figure 2b). Each electric discharge will generate a force whose direction is perpendicular to the cylindrical surface affected by electric discharges of the wire electrode. Regarding the maximum impulse force generated by a single electric discharge, Zhang et al. [35] established the empirical model *F* = 1.108*I*^0.416^, where *I* is the discharge current.

Analyzing the radial forces generated by the electric discharges in a section perpendicular to the wire axis, it is found that the components of the radial forces, perpendicular to the feed direction of the wire electrode in the test sample, will cancel each other out. In contrast, the components of the same forces acting along the direction of feed movement of the wire electrode will be able to deform the wire electrode. In this way, the wire electrode acquires a shape that has a delay (a vault) in the area from the approximate middle of the thickness of the test sample to the wire’s area at the wire’s entrance and exit electrode in the test sample.

This situation could highlight three distinct areas of the wire electrode: a curved area near the test sample and rectilinear areas for those parts of the wire electrode between the test sample and guide coils (Figure 3). The presence of the curvilinear area has been highlighted by various researchers ([1,6,10,13,15,16,19,28], etc.).

Let us consider the wire electrode as a string. The possible non-uniformity of the development of electric discharges on each of the quarters of the half-cylindrical surface of the wire electrode affected by the electric discharges (Figure 2b), but also other factors, will lead to an oscillation of the wire electrode perpendicular to the direction of feed movement of the wire electrode in the test sample.

Note that the occurrence of an oscillation of the wire electrode, for example, to the right, to the direction of the wire electrode entering the test sample, will create conditions for the development of electric discharges on about a quarter of the periphery of the circle corresponding to a cross-section through the wire electrode. The development of the electric discharges on a quarter of the wire electrode periphery (Figure 3b) will generate a force that will push the wire electrode in an approximately diametrically opposite direction, where the electric discharges will be primed on another quarter of the circular periphery corresponding to a cross-section through the wire electrode. The process may repeat itself, increasing the width of the slot generated in the test sample.

A consequence of the wire electrode oscillation is the increase of the evaluated slot width in a cross-section through the generated slot, at the level of the approximate middle area of the test sample, to the slot width at entering and exit of the tool electrode in the test sample. As seen in Figure 3, the wire oscillation in a plane perpendicular to the direction of feed movement of the wire in the plate-shaped test sample leads to a deviation from the flat shape of the slot walls, namely a deviation concave type. The size of this deviation could be evaluated by knowing the distinct values of the slot width in the area where the wire electrode enters the test sample, in the area corresponding to the middle of the test sample thickness, and in the area where the wire electrode leaves the test sample.

Several groups of factors influence the amplitude of the wire electrode oscillation. The main groups are the following:(a)Characteristics of electric discharges (voltage, current, pulse shape, pulse-on time, pulse-off time, filling factor, etc.)(b)Some physical-mechanical properties that define the behavior of the test sample material under the action of electric discharges (density, melting temperature, vaporization temperatures, etc.)(c)The characteristics of the dielectric liquid (density, fluidity, etc.) and its way of circulation in the working gap(d)The test sample thickness(e)The tension force of the wire electrode(f)The speed of movement of the wire electrode along its axis.

A brief analysis of the voltage in the wire electrode was performed using the finite element method. The analysis aimed to highlight the stress inside the electrode wire as it passes through the test sample (Figure 4). A type of static structural analysis with a beam model was chosen to represent the wire electrode with a circular cross-section. Nonlinear effects were allowed in using a beam instrument for which the maximum combined load was required. The section of the wire electrode, when it enters the test sample, is affected by significant stress because it is subjected to a force developed by electric discharges. Another type of result was obtained with the help of the wire electrode with a value for the sliding distance of 0.050812 mm as an effect of the sample test material pushing the wire in the opposite direction. The analysis is meant to provide a partial voltage distribution in the wire electrode, as occurs in the WEDM process.

Static structural analysis is a continuation of another type of analysis, namely thermal electric. The results of the first analysis were exported to the static structural analysis. The thermal electric analysis considered a distributed heat flux of Gaussian type or, more precisely, of Lorentzian–Gaussian type, applied in the work area. The hypothesis that the dielectric remains stable during the WEDM process was accepted [36,37].

### 2.2. Experimental Conditions

The main objective pursued by the experimental research was to highlight the influence exerted by various factors on the variation of the width of the slot generated in the test sample by wire electrical discharge machining in a cross-section through the slot generated in the test sample.

Experimental tests on the influence of input factors in the WEDM process on the flatness error were performed on a Japax L250 (made in Japan). These experimental tests were part of a larger set of research in which, as output parameters, the roughness of the processed surfaces, the wire electrode wear, the material removal rate, etc., were still taken into account. A wire electrode made of copper with a diameter of 0.2 mm was used. Slots were thus made in plate-type test samples made of alloy steel (2 % C, 0.40 % Mn, 0.30 % Si, 12 % Cr, 0.35 % Ni) with different thicknesses.

To avoid the possible special behavior of the wire at the entrance to the test sample, after the application of WEDM, an abrasion removal of a layer of test sample material with a thickness of approximately 2–3 mm was performed. (Figure 5). Subsequently, the widths *a*, *m*, and *b* of the slot made by WEDM were measured at the input of the wire electrode in the test sample, in the middle area of the test sample thickness, and at the exit of the wire electrode in the test sample.

When determining the number of experimental tests, 7 input factors with values on two levels according to the Taguchi methodology were initially considered. The 7 factors were: test sample thickness *H*, pulse-on time *T_on_*, pulse-off time *T_off_*, the current intensity *I*, wire traveling speed *v*, tension force *F* applied to the wire electrode, and the average voltage *U*, respectively.

Subsequently, it was considered that the average voltage of the electric discharge could also influence the slot widths taken into account for evaluating deviations from the flat shape of the surfaces generated by WEDM. For this reason, 16 experimental tests were performed for 4 distinct values of the average voltage of electric discharges.

Measuring the values of the size of the gap *a* at the entrance of the wire electrode in the test sample (at the bottom of the test sample), of the size *b* of the gap at the exit of the wire electrode from the test sample (at the top of the test sample), and in an area corresponding approximately to the middle of the test sample thickness was performed using a Carl Zeiss optical microscope of type BK70 × 50. The measurement of each size was repeated 3 times. Table 1 included the measured slot widths and the average value of the slot width corresponding to a certain experiment.

Table 1 lists the values of the input factors in the proposed WEDM process, and the results obtained by measurement (*a_med_*, *b_med_*, and *m_med_* values).

Figure 5b shows the slot widths taken into account as output parameters of the studied process, namely:*a*—the width of the slot in the lower part of the test sample, where the wire electrode enters the test sample*b*—the width of the slot in the upper area of the test sample, where the wire electrode comes out of the test sample*m*—the width of the slot corresponding approximately to the middle of the thickness *H* of the test sample.

As values of process input factors that correspond to the two variation levels, the following were used:-test sample thickness *H*_1_ = 1 mm and *H*_2_ = 20 mm-pulse-on time *T_on_*
_1_ = 2 µs and *T_on_*
_2_ = 12 µs-pulse-off time *T_off_*
_1_ = 7 µs and *T_off_*
_2_ = 30 µs-average current intensity *I*_1_ = 3.1 A and *I*_2_ = 3.8 A-tension force *F*_1_ = 2.45 N and *F*_2_ = 10.54 N-wire movement speed along with its axis *v*_1_ = 500 mm/min and *v*_2_ = 2500 mm/min-average voltage *U*_1_ = 135 V and *U*_2_ = 160 V.

When establishing the values of some input factors in the WEDM process, the recommendations of the manufacturer of machine to machine by electrical discharges with wire electrodes were taken into account.

## 3. Results

The conditions for performing the experimental tests and the results of these tests are listed in Table 1. Thus, in columns 2–8, the values of the input factors *H*, *T_on_*, *T_off_*, *I*, *F*, *v*, and *U* were included in each of the 64 experimental trials.

## 4. Discussion

To process the experimental results to identify empirical mathematical models that would highlight the influence exerted by the values of the input factors on the quantities that define the slot widths, software based on the least-squares method [38] was used. This software selects the most appropriate mathematical model for the experimental results between five mathematical models (first-degree polynomial, second-degree polynomial, power-type function, logarithmic function, and hyperbolic function). It was estimated that for the intervals of variation of the magnitudes of the input factors, there could be a monotonous variation of the magnitude of the output parameters to consider. In this way, power-type mathematical models have been preferred, as such models are quite often used in the study of machining processes (for example, in the Taylor model of the influences exerted by various factors on the cutting tool life, in some models valid for cutting forces, for some parameters that characterize the roughness of surfaces obtained by machining, etc.).

The value of Gauss’s criterion was used to evaluate the adequacy of the established empirical mathematical models to the experimental results. The Gauss criterion value is defined using the sum of the squares of the differences between the measured values and those obtained using the empirical mathematical model considered. The lower the value of Gauss’s criterion, the more appropriate the empirical mathematical model applied is considered.

Taking into account the possibilities of the software used, the most appropriate mathematical models for the experimental results obtained were determined from the five types of mathematical models available. Power function-type mathematical models were also determined. For each of these models, the value of Gauss’s criterion was also specified to highlight the degree of adequacy of the proposed empirical mathematical models to the experimental results obtained.

It was thus found that among the five empirical mathematical models possible to be used, only in the case of the *b_med_* width of the slot was the most appropriate empirical mathematical model a second-degree polynomial:(2)$$\begin{array}{l} b_{med}=-162.3186-1.0796H-0.01624H_{}^{2}-35.3780T_{on}+2.7371T_{on}^{2}\\ \;\;\;\;\;\;\;\;\;\;\;\;\;\;\;\;\;\;\;\;-34.0689T_{off}+0.8864T_{off}^{2}+161.1226I-13.5867I_{}^{2}\\ \;\;\;\;\;\;\;\;\;\;\;\;\;\;\;\;\;\;\;\;+73.5865F-6.6565F^{2}+0.7031v^{}-0.00023v^{2}\\ \;\;\;\;\;\;\;\;\;\;\;\;\;\;\;\;\;\;\;\;-1.5818U_{}-0.00765U_{}^{2} \end{array}$$

-for which Gauss’s criterion has the value *S_G_* = 2354.82

In the other two cases, namely for the widths *a_med_* and *m_med_*, it was observed that the empirical mathematical models of power-type function proved the most adequate for the experimental results obtained, so that, finally, these mathematical models were accepted to highlight the influence exerted by the input factors in the investigated process on the widths of the slots taken into account:-for the slot width *a_med_*:
(3)$$a_{med}=19.9113H_{}^{0.0117}T_{on}^{0.0355}T_{off}^{-0.0615}I_{}^{0.5953}F^{-0.2026}v^{-0.0108}U_{}^{0.497}$$
for which the Gauss’s criterion has the value *S_G_* = 2337.5-for the slot width *m_med_*:
(4)$$m_{med}=16.8687H_{}^{0.0126}T_{on}^{0.0348}T_{off}^{-0.0607}I_{}^{0.585}F^{-0.2024}v^{-0.009}U_{}^{0.5301}$$
the Gauss’s criterion having the value *S_G_* = 2319.72-for the slot width *b_med_*:
(5)$$b_{med}=31.865H_{}^{0.0068}T_{on}^{0.036}T_{off}^{-0.0643}I_{}^{0.5953}F^{-0.21}v^{-0.0107}U_{}^{0.404}$$-for which the Gauss’s criterion has the value *S_G_* = 2359.49

To graphically highlight the influences exerted by some of the input factors considered on the values of slot widths *a_med_*, *m_med_* and *b_med_*, the graphical representations included in Figure 6, Figure 7, Figure 8, Figure 9 and Figure 10 were elaborated.

The analysis of empirical mathematical models and graphical representations facilitated the formulation of observations regarding the objectives pursued by experimental research.

An examination of the three empirical mathematical models constituted by Equations (3)–(5) shows, first of all, the relatively close values of the exponents attached to each of the seven input factors considered, which presupposes a similar influence exerted by these factors on the widths of the slits determined in the upper and lower part of the test piece, as well as at the middle of the test sample thickness.

On the other hand, it was found that an increase in the values of the test sample thickness *H*, pulse on-time *T_on_*, average current *I*, wire traveling speed *v*, and average working voltage *U* causes an increase in the values of the parameters *a*, *m*, and *b*, since the values of the exponents attached to these factors in all the three empirical mathematical models have positive values. In addition, an increase in the values of the pulse-off time T_*off*_, wire tension *F*, and wire traveling speed *v* causes a decrease in the average values of the parameters *a*, *m*, and *b*, as the values of the exponents attached to these input factors are negative.

It was found that the average current exerts the strongest influence on the quantities *a*, *m*, and *b* since, in the empirical mathematical models, this factor was associated with the highest absolute values of the exponents.

The graphical representation in Figure 7 confirms the larger value of the width of the slot *m* in the middle of the thickness of the test sample. Still, a value close to this maximum value of *m* also corresponds to the area where the wire electrode moving along its axis enters the test piece.

Another factor influencing the slots’ widths *a*, *m*, and *b* is the average voltage *U* (Figure 7). Although both the average current *I* and the average voltage *U* have been considered input factors, there is likely a dependence between them, the variation of the current intensity being determined by the variation of the voltage *U*.

The increase of the average voltage *U* and the average current *I* lead to the increase of the energy of the electric discharges, which means an increase in the working gap and, therefore, an increase in the width of the slot made in the test sample.

The third factor in terms of the influence exerted on the width of the slot is the wire tension *F*, whose exponents in empirical mathematical models have values of 0.20–0.21. As expected, an increase in the wire tension leads to a decrease in the width of the slot by decreasing the amplitude of the oscillations of the wire electrode (Figure 8).

Regarding the influence exerted by the wire tension *F* on the width of the slot, Figure 8 shows that there are very small differences between the widths of the slot at the entrance of the wire electrode in the test piece and, respectively, in the middle of the thickness *H* of the test sample, as in the graphical representation, the curves corresponding to *a* and *m* overlap and on the diagram, it is not easy to notice the difference between the values of the points that contribute to the composition of the curves.

In the diagram in Figure 9, the variation of the width *m* of the slot corresponding to the middle of the thickness *H* of the test sample depending on the average current *I* and the average voltage *F* is highlighted. The increase of the width *m* of the width when the current *I* increases and respectively the decrease of the width *m* when the wire tension *F* increases can be observed.

An image of the variation of the slot width *m_med_* as a function of both the average current *I* and the tension force *F* can be seen in the diagram in Figure 10. It can be seen that, as expected, for a certain value of the tension force, the increase of the average cur-rent *I* leads to an increase in the width of the *m_med_*, due to the increase in the energy of electric discharges. At the same time, for a certain value of the average current *I*, when the tension force *F* increases, there will be a decrease in the *m_med_* width of the slot due to the decrease of the vibration amplitude of the tool electrode.

A possible explanation of the larger value of the width of the slot where the wire electrode enters the test sample (at the bottom of the plate type test sample) to the value *b* of the width at the exit of the slot from the test sample (Figure 5) can be formulated by considering that if some metal particles detached from the test sample move down under the action of gravity, they could cause the generation of spurious electrical discharges that will contribute to an increase in the width of the slot.

## 5. Conclusions

The width of the slot generated by WEDM can be taken into account in the area of entry of the wire electrode into the test sample, in the area of exit of the wire electrode from the test sample, and respectively in the area corresponding to the middle of the thickness of the test sample.Distinct factors influence the slot width values in the three distinct areas mentioned. A factorial experiment was designed and materialized, with six independent variables at two levels of variation and one variable at four levels of variation; The seven independent variables considered were the thickness of the test sample, the pulse-on time, the pulse-off time, the average current, the wire electrode tension force, the wire traveling speed, and the average voltage.By mathematically processing the experimental results, empirical models of power type were established for the three widths of the slot. Empirical mathematical models highlight the variation of slot widths and the intensity of the influence exerted by the seven factors taken into account on the values of slot widths.The values of the exponents attached to the input factors in the WEDM process were examined, providing information on the intensity of the influence exerted by each of the input factors in the WEDM process on the width of the slot in the three areas.It was found that the strongest influence on the width of the slot is exerted by the average current, followed by the average voltage and the tension force of the wire. The other factors exert a low or negligible influence on the variation of the slot width generated by WEDM.In the future, there is the intention to perform experimental tests on test samples made of other materials, following the influence of the seven factors on other output parameters of the WEDM process, such as the roughness of the machined surface, the wear of the wire electrode, and the material removal rate.

## Figures and Tables

**Figure 1 micromachines-13-01231-f001:**
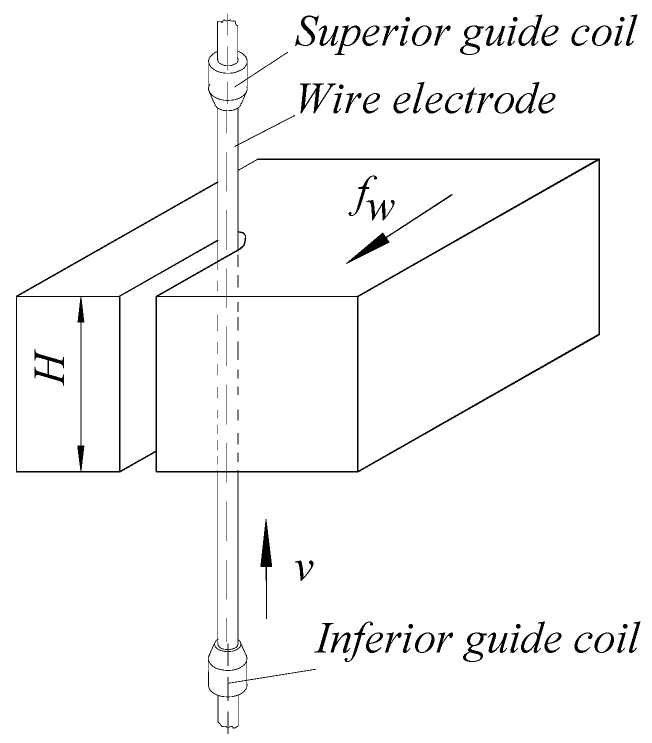
Gradual development of a slot by electrical discharge machining using a wire electrode supported in two guide coils.

**Figure 2 micromachines-13-01231-f002:**
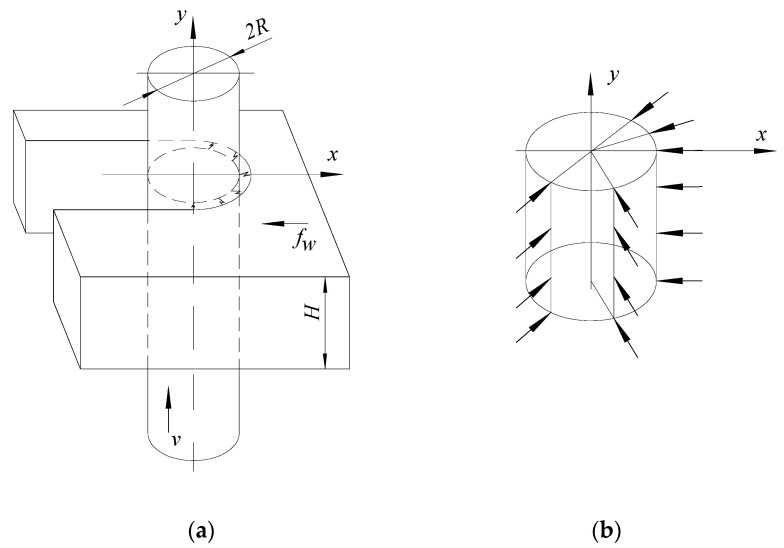
Generation of electric discharges only on half of the cylindrical surface of the wire electrode in the workpiece: (**a**)—position of the wire in the slot; (**b**)—electric discharges acting on half of the cylindrical surface of the wire electrode.

**Figure 3 micromachines-13-01231-f003:**
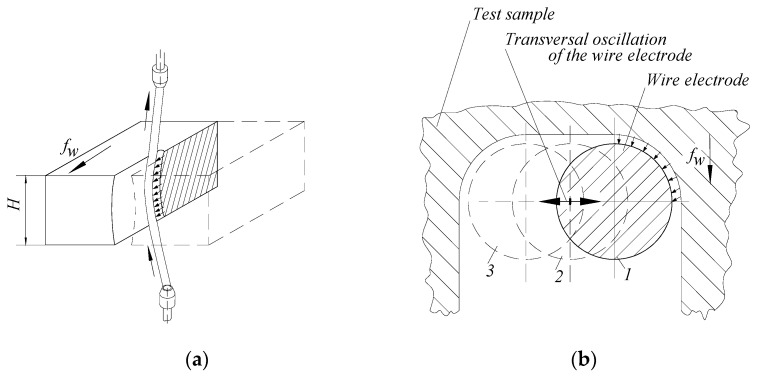
Lag of the wire electrode in the middle area of the test sample (**a**) and a cross-section through the middle area of a test sample to observe the oscillation of the wire electrode in a plane perpendicular to the direction of entry of the wire electrode in the test sample, in the middle area of the test sample (**b**): 1—position of the electrode wire at the end of the oscillation to the right; 2—the central position of the wire electrode; 3—wire electrode position at the end of the oscillation to the left.

**Figure 4 micromachines-13-01231-f004:**
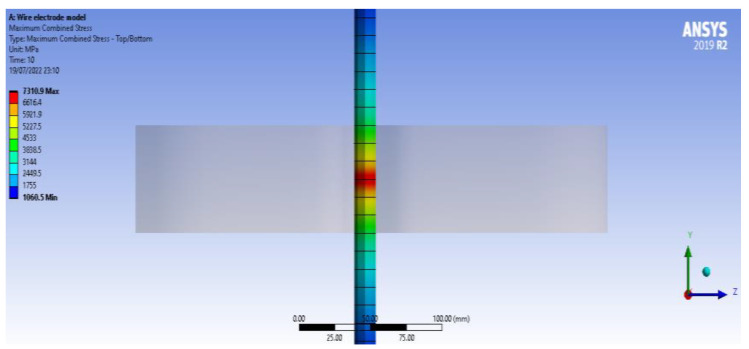
Stress in the wire electrode when it passes through the test sample due to the force generated by the electrical discharges.

**Figure 5 micromachines-13-01231-f005:**
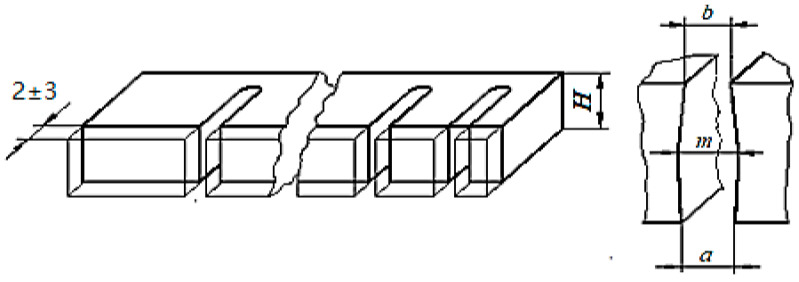
Making short slots in the test sample by wire electrical discharge machining in the experimental tests (**a**) and the main shape of the cross-section through the machined slot (**b**).

**Figure 6 micromachines-13-01231-f006:**
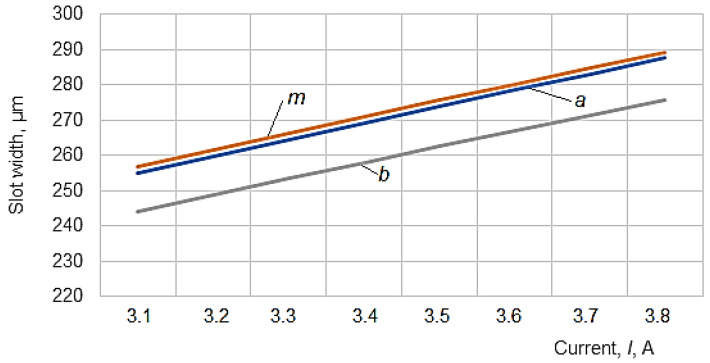
Influence of current *I* on the widths of the slots when the wire electrode enters the test sample (*a*), in the middle of the thickness of the test sample (*m*), and at the exit of the wire electrode in the test sample (*b*) (*H* = 20 mm, *T_on_* = 12 µs, *T_off_* = 30 µs, *F* = 10.54 N, *v* = 1500 mm/min, *U* = 160 V).

**Figure 7 micromachines-13-01231-f007:**
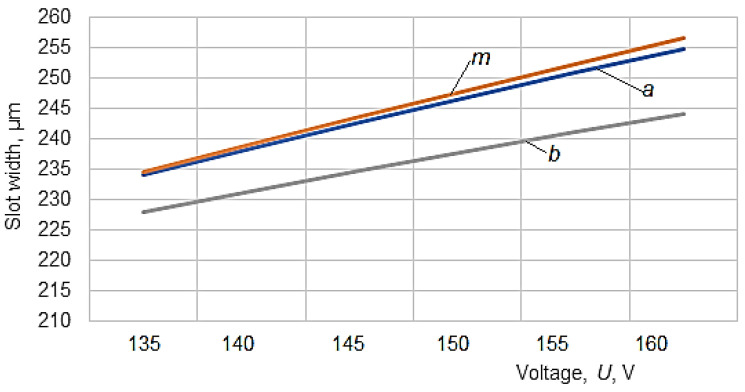
Influence of voltage *U* on the slot width entering the wire electrode in the test sample (*a*), in the middle of the thickness of the test sample (*m*), and at the exit of the wire electrode in the test sample (*b*), respectively (*H* = 20 mm, *T_on_* = 12 µs, *T_off_* = 30 µs, *I* = 3.1 A, *F* = 10.54 N, *v* = 2500 mm/min).

**Figure 8 micromachines-13-01231-f008:**
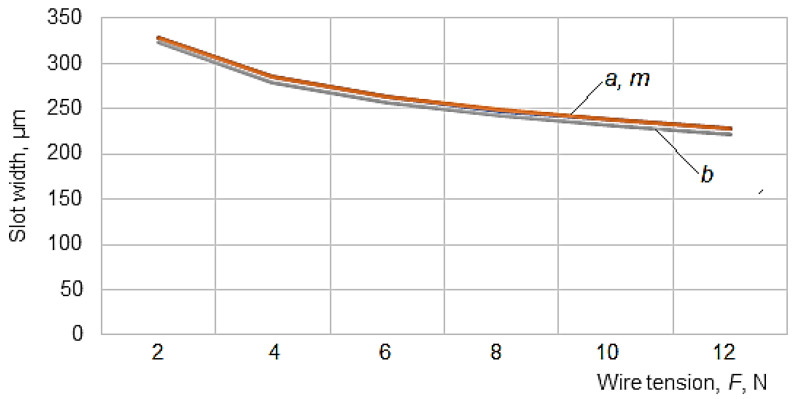
Influence of the wire tension *F* on the width of the slot when the wire electrode enters the test sample (*a*), in the middle of the thickness of the test sample (*m*), and at the exit of the wire electrode from the test sample (*b*) (*H* = 20 mm, *T_on_* = 12 µs, *T_off_* = 30 µs, *I* = 3.1 A, *F* = 10.54 N, *v* = 2500 mm/min).

**Figure 9 micromachines-13-01231-f009:**
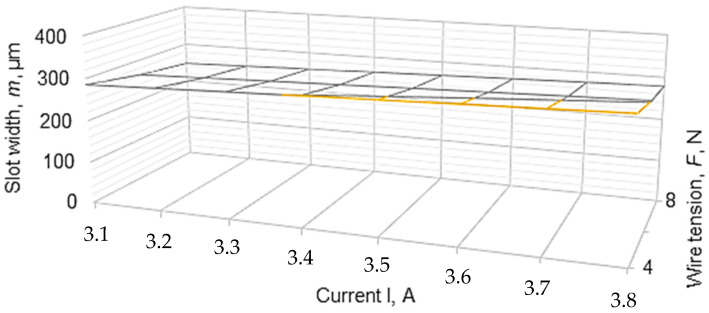
The influence of the average current *I* and the average voltage *F* on the width of the slot in the middle of the thickness of the blank (*H* = 20 mm, *T_on_* = 12 µs, *T_off_* = 30 µs, *v* = 2500 mm/min, *U* = 135 V).

**Figure 10 micromachines-13-01231-f010:**
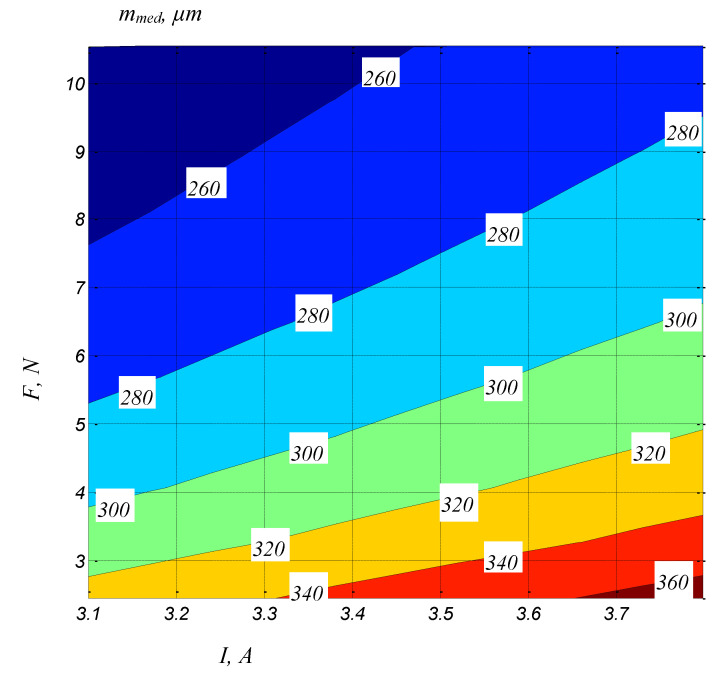
The influence of the average current *I* and the average voltage *F* on the width of the slot in the middle of the thickness of the blank (*H* = 20 mm, *T_on_* = 12 µs, *T_off_* = 30 µs, *v* = 2500 mm/min, *U* = 135 V).

**Table 1 micromachines-13-01231-t001:** Experimental conditions and results.

	Input Factors							Output Parameters					
Experiment No.	*H*, mm	*T_on_*, μs	*T_off_*, μs	*I*, A	*F*, N	*v*, mm/min	*U*, V	Measured Values for *b*, μm	*b_med_*, μm	Measured Values for *m*, μm	m_med_, μm	Measured Values for *a*, μm	*a_med_*, µm
Column1	2	3	4	5	6	7	8	9	10	11	12	13	14
1	1	2	7	3.1	2.45	500	135	281,282,287	283	281,282,287	283	281,282,287	283
2	1	2	7	3.8	10.54	2500	135	270,273,269	270	270,273,269	270	270,273,269	270
3	1	2	30	3.1	2.45	2500	135	265,261,271	265	265,261,271	265	265,261,271	265
4	1	2	30	3.8	10.54	500	135	287,281,283	283	287,281,283	283	287,281,283	283
5	1	12	7	3.8	2.45	2500	135	315,289,298	300	315,289,298	300	315,289,298	300
6	1	12	7	3.1	10.54	500	135	254,259,258	257	254,259,258	257	254,259,258	257
7	1	12	30	3.1	2.45	500	135	312,306,320	312	312,306,320	312	312,306,320	312
8	1	12	30	3.8	10.54	2500	135	268,273,269	270	268,273,269	270	268,273,269	270
9	20	2	7	3.1	2.45	2500	135	280,291,288	286	294,288,293	291	304,282,285	290
10	20	2	7	3.8	10.54	500	135	302,298,300	300	300,306,311	305	302,303,309	304
11	20	2	30	3.1	2.45	500	135	296,291,293	293	302,297,299	299	288,290,315	297
12	20	2	30	3.8	10.54	2500	135	250,255,252	252	259,254,261	258	254,249,267	256
13	20	12	7	3.8	2.45	500	135	381,395,375	383	393,387,389	389	388,380,396	388
14	20	12	7	3.1	10.54	2500	135	258,251,250	253	256,259,260	258	257,254,260	257
15	20	12	30	3.1	2.45	2500	135	356,351,351	352	353,359,363	358	357,356,358	357
16	20	12	30	3.8	10.54	500	135	230,234,242	235	245,238,239	240	236,248,235	239
17	20	2	7	3.1	2.45	500	160	300,301,307	302	310,315,314	313	311,304,312	309
18	20	2	7	3.8	10.54	2500	160	332,330,328	330	338,343,340	340	332,339,340	337
19	20	2	30	3.1	2.45	2500	160	301,306,305	304	315,311,316	314	305,315,310	310
20	20	2	30	3.8	10.54	500	160	278,272,272	274	282,284,286	284	284,273,286	281
21	20	12	7	3.8	2.45	2500	160	405,400,398	401	415,410,412	412	400,416,408	408
22	20	12	7	3.1	10.54	500	160	287,292,291	290	302,299,298	299	293,283,315	297
23	20	12	30	3.1	2.45	500	160	381,384,384	383	392,396,394	394	391,386,393	390
24	20	12	30	3.8	10.54	2500	160	269,267,267	267	279,278,276	277	271,278,273	274
25	40	2	7	3.1	2.45	2500	160	316,312,306	311	325,323,322	323	326,314,320	320
26	40	2	7	3.8	10.54	500	160	328,327,325	326	336,339,341	338	324,338,346	336
27	40	2	30	3.1	2.45	500	160	308,311,315	311	320,326,325	323	312,326,322	320
28	40	2	30	3.8	10.54	2500	160	239,242,242	241	256,252,252	253	249,253,248	250
29	40	12	7	3.8	2.45	500	160	431,428,427	428	441,442,443	442	427,440,447	438
30	40	12	7	3.1	10.54	2500	160	245,240,239	241	250,252,259	253	253,247,250	250
31	40	12	30	3.1	2.45	2500	160	411,409,407	409	423,422,418	421	423,414,417	418
32	40	12	30	3.8	10.54	500	160	236,232,231	233	246,243,244	244	239,247,240	242
33	40	2	7	3.1	2.45	500	190	310,308,306	308	329,326,325	326	326,316,318	320
34	40	2	7	3.8	10.54	2500	190	364,364,367	365	384,385,382	383	376,381,377	378
35	40	2	30	3.1	2.45	2500	190	339,340,337	338	356,354,360	356	356,342,352	350
36	40	2	30	3.8	10.54	500	190	335,339,337	337	352,355,357	354	359,342,349	350
37	40	12	7	3.8	2.45	2500	190	446,448,447	447	464,469,468	467	467,453,460	460
38	40	12	7	3.1	10.54	500	190	316,313,310	313	333,334,324	330	326,319,333	326
39	40	12	30	3.1	2.45	500	190	390,386,385	387	408,408,404	406	396,405,399	400
40	40	12	30	3.8	10.54	2500	190	263,266,267	265	281,283,285	283	274,268,289	277
41	60	2	7	3.1	2.45	2500	190	387,385,383	385	401,405,414	406	406,397,400	401
42	60	2	7	3.8	10.54	500	190	370,374,367	370	393,392,391	392	395,381,388	388
43	60	2	30	3.1	2.45	500	190	330,325,329	328	350,351,349	350	339,350,343	344
44	60	2	30	3.8	10.54	2500	190	234,230,226	230	256,252,249	252	249,240,252	247
45	60	12	7	3.8	2.45	500	190	474,471,470	471	492,496,497	495	487,492,488	489
46	60	12	7	3.1	10.54	2500	190	225,228,222	225	246,248,246	246	245,238,240	241
47	60	12	30	3.1	2.45	2500	190	460,464,466	463	486,485,483	484	485,476,479	480
48	60	12	30	3.8	10.54	500	190	233,229,226	229	251,248,250	249	249,240,249	246
49	60	2	7	3.1	2.45	500	215	320,325,318	321	349,352,349	350	343,338,339	340
50	60	2	7	3.8	10.54	2500	215	410,419,413	414	440,447,439	442	430,433,439	434
51	60	2	30	3.1	2.45	2500	215	370,373,373	372	406,399,395	400	394,387,389	390
52	60	2	30	3.8	10.54	500	215	380,373,381	378	400,397,417	404	396,399,399	398
53	60	12	7	3.8	2.45	2500	215	515,518,521	518	547,542,558	549	539,542,536	539
54	60	12	7	3.1	10.54	500	215	342,349,335	342	362,369,374	368	360,365,361	362
55	60	12	30	3.1	2.45	500	215	389,395,407	397	419,427,434	426	413,419,419	417
56	60	12	30	3.8	10.54	2500	215	263,269,254	262	293,288,286	289	284,277,279	280
57	80	2	7	3.1	2.45	2500	215	340,337,328	335	369,366,369	368	364,358,358	360
58	80	2	7	3.8	10.54	500	215	330,339,330	333	361,365,370	365	363,359,358	360
59	80	2	30	3.1	2.45	500	215	345,348,342	345	372,379,385	378	367,374,369	370
60	80	2	30	3.8	10.54	2500	215	226,220,226	224	259,258,255	257	252,248,250	250
61	80	12	7	3.8	2.45	500	215	508,516,509	511	543,547,550	546	536,541,537	538
62	80	12	7	3.1	10.54	2500	215	325,329,321	325	353,359,362	358	352,346,352	350
63	80	12	30	3.1	2.45	2500	215	510,518,520	516	546,549,551	548	544,539,543	542
64	80	12	30	3.8	10.54	500	215	228,221,223	224	251,257,256	254	253,247,250	250

## Data Availability

The data presented in this study are available in https://drive.google.com/drive/folders/1G_eCNuxYsk88kx_OdIWS2kzbgzytLJEP?usp=sharing (accessed on 11 July 2022).

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
