# Peer review of "WEDM-Generated Slot Width Variation Modeling"

_micromachines, 2022, doi:10.3390/mi13081231_

Round 1
Reviewer 1 Report
Yes, I agree with the authors of the manuscript that geometry shape errors in WEDM are a serious problem that requires several finishing cuts to be made afterwards. At the same time, this deficiency prevents WEDM technology from competing with other conventional but also progressive machining technologies in terms of process performance. Therefore, from this point of view, I consider experimental research to be very relevant. At the same time, I consider the information obtained from experimental research to be valuable.
Although the manuscript is well written, some corrections need to be made and the following questions answered:
1. It is generally known that the size of the width of the cutting gap depends on the combination of the setting of the main technological parameters, of which Ton, Toff, I, F, v, and U have a significant position. When evaluating the influence of individual input technological parameters of the WEDM process on the width of the cutting gaps, it is necessary for the values ​​of the other parameters to be constant during the machining process. Were these conditions met during the experiment? What was the range of these parameters. It is necessary to add them to the manuscript in tabular form.
2. There are many known facts in the introduction. It is necessary to reduce them.
3. It is necessary to more precisely specify the used material of the experimental samples, tool electrode, including their marking in accordance with international standards. What electroerosive device was used. Necessary to specify in the manuscript.
4. Are the stated results of experimental findings applicable to the machining of other materials using the given cutting tools and machining conditions? It should be discussed at the end of the manuscript.
5. At the end of the thesis, there is no discussion in which the authors evaluate the conclusions in the context of existing knowledge. It is very important to highlight points of agreement or disagreement between the results in this paper and the cited references in the manuscript, either qualitatively or quantitatively, in the conclusion of the thesis.
The paper is prepared at a good level and after modifications and additions can be published in the journal Micromachine.
Author Response
Authors' responses to the reviewers' comments
First of all, the authors of the reviewed paper wish to express their gratitude for the efforts of the reviewers invested in the analysis of the proposed paper and for the useful observations and suggestions to improve the quality of the paper.
All the changes were highlighted in the manuscript of the paper by using the color green.
REVIEWER 1
Reviewer's comment no. 1. The authors conducted a large review of the data in the Background section. And at the end of the section, it concludes "However, it was found that there is relatively little information on the variation of the slot width generated by WEDM." In my opinion, the authors artificially wrote such a conclusion. The authors should pay attention to the works "Fundamental geometry analysis of wire electrical discharge machining in corner cutting" by W. J. Hsue, Y. Liao, Shui-Shong Lu Published 1 April 1999. Study the article Ultrasonic Vibration Assisted Electro-Discharge Machining (EDM)—An Overview https://doi.org/10.3390/ma12030522. Study the work of Vibration modeling and analysis of wire during the WEDM process -DOI:10.1080/10910344.2015.1085312. It is also worth studying the works of scientists from Russia working in this direction, the authors: Ablyaz, Fedorov, Volgin and others. Works and authors submitted for study should be included in the References.
Thus, after analyzing these works, it is necessary to correct the article. It is worth refraining from categorical conclusions. It is necessary to show your unique amount of experimental data. But it is not necessary to say that the issue has not been fully studied. This is not true.
Authors' response to the reviewer's comment. The authors considered the reviewer's suggestions, included some of the suggested papers in the list of references, and included references in the text of the manuscript.
However, we would like to mention that:
- In Hsue et al. [6], there is no reference to the width of the slot made by the WEDM process;
- In the work Sabyrov et al. [27], there is only a very general reference to obtaining slots by WEDM (“The UV assisted EDM can also be useful in machining large 3D structures using conventional die-sinking EDM and high aspect ratio and thin slot cutting using wire EDM. ”) And no reference to the variation of the slot width obtained by WEDM;
- The work Ablyaz and Lesnikov (Influence of the Wire EDM Conditions on the Cut Width) [18] does refer to the width of the slot but measured only from above the slot. We found the reviewer's suggestion useful, and Ablyaz and Lesnikov's work was included in the list of bibliographic references;
- There are works on the web with A.A. Fedorov as the first author, but these papers refer to other aspects of WEDM, not addressing issues regarding the variation of slot width in a section perpendicular to the slot;
- The works produced by V. Volgin as first author or co-author, works existing on the web, do not seem to have addressed problems regarding the variation of the width of the slot in a section perpendicular to the slot.
In these conditions, we consider that the wording used in the manuscript (“However, it was found that there is relatively little information on the variation of the slot width generated by WEDM”) is not categorical. The authors of the manuscript acknowledge that it is possible that works regarding the variation of the slot width generated by WEDM, works that are not known to the authors of the manuscript. A categorical statement would have been: "However, it was found that there is no information on the variation of the slot generated by WEDM."
Reviewer's comment no. 2. You need to change the title of the article. Shift the focus to empirical modeling. You write about Shape errors of the surfaces, but in fact you are studying the width of the slot. These are different concepts. It is worth considering in more detail the indicators of Shape errors of the surfaces - for example, flatness deviation. After that change the conclusions.
Authors response to the reviewer's comment. The authors considered that the reviewer was right. The article's title has been changed to "WEDM-generated slot width variation modeling".
Reviewer's comment no. 3. The conclusions must be redone. No specifics. General words. Show dependencies. Show the uniqueness of your work. Describe physical processes.
Authors response to the reviewer's comment. The authors modified the article's conclusions, in line with the reviewer's suggestions.
Reviewer's comment no. 4. Pictures 1-5 for what in the article? This is a repetition of traditional schemes and drawings not invented by you. Why are they at work? What is their uniqueness?
Authors response to the reviewer's comment. Figures 1-4 were introduced in the paper to highlight that the oscillation of the wire electrode leads to a variation of the slot width generated by WEDM in the test sample and also to facilitate the understanding of how different factors influence the variation of the measured slot width in the area where the wire electrode enters the test sample, in the area where the wire electrode get out of the test sample and in the middle area of the thickness of the test sample. The figures are not simple reproductions of some schemes encountered in the accessed literature, in which case it would have been necessary to obtain the copyright document.
Reviewer's comment no. 5. The article needs serious revision. For publication in a journal of this level, it is worth making significant changes. At the moment, the work looks like extended conference paper.
Authors response to the reviewer's comment. The authors took the reviewer's comments into account and made changes to the manuscript.

Reviewer 2 Report
1. In figures, some symbols are not clearly defined. For example, fw.
2. Please adjust the location of reference 19.
3. Please check the definition of U in manuscript.
4.The authors declared that “Figure 8 shows that there are very small differences between the widths of the slot at the entering of the wire electrode in the test piece and, respectively, in the middle of the thickness H of the test sample …”. Why?
5.The Conclusion should be more informative the experimental data.
Author Response
Authors' responses to the reviewers' comments
First of all, the authors of the reviewed paper wish to express their gratitude for the efforts of the reviewers invested in the analysis of the proposed paper and for the useful observations and suggestions to improve the quality of the paper.
All the changes were highlighted in the manuscript of the paper by using the color green.
REVIEWER 2
Reviewer's comment no. 1. Figure 3 and 4 can be only one. Please consider it .
Authors' response to the reviewer's comment. The authors appreciated that the reviewer was right. As the contents of the two figures were close, figure 3 was abandoned.
Reviewer's comment no. 2. Regarding the importance of the problem, WEDM errors in shapes is an old problem, so the discussion must include the first Works about it, Computer simulation of wire-EDM taper-cutting, International Journal of Computer Integrated Manufacturing 19 (7), 727-735 and new ones as the classic Multi-response optimization of WEDM process parameters for machining of superelastic nitinol shape-memory alloy using a heat-transfer search algorithm, Materials 12 (8), 1277. You can improve your theoretical 2.2 section with them.
Authors response to the reviewer's comment. The authors appreciated that the reviewer was right and included a reference to the work of Chaudhary et al. [31].
Reviewer's comment no. 3. Forces in WEDM are due to different causes. Ref 11 is not bad, but above and below were real references in the 200s..
Authors response to the reviewer's comment. Considering that the reviewer is right, the manuscript's authors extended the list of references to the forces acting on the filiform electrode.
Reviewer's comment no. 4. Conclusions are weak, much better as separated points, one per each main contribution.
Authors response to the reviewer's comment. We thought the reviewer was right. The conclusions of the paper have been reformulated.
Reviewer's comment no. 5. Are you sure your machine is a Japax L250? It is the first news about this brand.
Authors response to the reviewer's comment. We will try to prove the existence of the Japax L250 by including a copy of JAPAX Technical Information No. 103.
The copy was made after:
Reviewer's comment no. 6. Journal of materials processing technology 182 (1-3), 574-579 gave also a key to reduce eeros in corners, even though the process is characterised by its high accuracy level (sufficient even for micromachining applications), the development of enhanced generators that produce more energetic discharges yielding cutting speeds as high as 500 mm2/min has resulted in stronger forces acting on the wire. These forces, together with the low rigidity of the wire, especially in the cutting of parts of high thickness, are responsible for wire deformation that has a direct influence on the accuracy of the part, mainly on wall-flatness and corners
Authors response to the reviewer's comment. The authors considered the reviewers' commentary and included references to three of the works authored by J.A. Sanchez, which address issues related to the accuracy of the WEDM process.
Reviewer's comment no. 7. Paper can only be accepted if the discussion is better, based on missed works with high quality machines
Authors' response to the reviewer's comment. The authors took into account the reviewer's comment.

Reviewer 3 Report
The development of the technology of WEDM is an important scientific and technological task.
The authors pay attention to the study of the width of the slot depending on various factors.
The processing accuracy depends on this parameter. I fully agree with the need for research in this area.
However, I must highlight the shortcomings in the work:
1. The authors conducted a large review of the data in the Background section. And at the end of the section, it concludes "However, it was found that there is relatively little information on the variation of the slot width generated by WEDM." In my opinion, the authors artificially wrote such a conclusion. The authors should pay attention to the works "Fundamental geometry analysis of wire electrical discharge machining in corner cutting" by W. J. Hsue, Y. Liao, Shui-Shong Lu Published 1 April 1999. Study the article Ultrasonic Vibration Assisted Electro-Discharge Machining (EDM)—An Overview https://doi.org/10.3390/ma12030522. Study the work of Vibration modeling and analysis of wire during the WEDM process -DOI:10.1080/10910344.2015.1085312. It is also worth studying the works of scientists from Russia working in this direction, the authors: Ablyaz, Fedorov, Volgin and others. Works and authors submitted for study should be included in the References.
Thus, after analyzing these works, it is necessary to correct the article. It is worth refraining from categorical conclusions. It is necessary to show your unique amount of experimental data. But it is not necessary to say that the issue has not been fully studied. This is not true.
2. You need to change the title of the article. Shift the focus to empirical modeling. You write about Shape errors of the surfaces, but in fact you are studying the width of the slot. These are different concepts. It is worth considering in more detail the indicators of Shape errors of the surfaces - for example, flatness deviation. After that change the conclusions.
3. The conclusions must be redone. No specifics. General words. Show dependencies. Show the uniqueness of your work. Describe physical processes.
4. Pictures 1-5 for what in the article? This is a repetition of traditional schemes and drawings not invented by you. Why are they at work? What is their uniqueness?
The article needs serious revision. For publication in a journal of this level, it is worth making significant changes. At the moment, the work looks like extended conference paper.
Author Response
Please see the document attached.

Reviewer 4 Report
A classic problem that can be helped with the new use of algorithms, see below.
Figure 3 and 4 can be only one. Please consider it.
· Regarding the importance of the problem, WEDM errors in shapes is an old problem, so the discussion must include the first Works about it, Computer simulation of wire-EDM taper-cutting, International Journal of Computer Integrated Manufacturing 19 (7), 727-735 and new ones as the classic Multi-response optimization of WEDM process parameters for machining of superelastic nitinol shape-memory alloy using a heat-transfer search algorithm, Materials 12 (8), 1277. You can improve your theoretical 2.2 section with them.
· Forces in WEDM are due to different causes. Ref 11 is not bad, but above and below were real references in the 200s.
· Conclusions are weak, much better as separated points, one per each main contribution.
· Results and regression model is OK, the set of values are modern.
· Are you sure your machine is a Japax L250? It is the first news about this brand.
· Journal of materials processing technology 182 (1-3), 574-579 gave also a key to reduce eeros in corners, even though the process is characterised by its high accuracy level (sufficient even for micromachining applications), the development of enhanced generators that produce more energetic discharges yielding cutting speeds as high as 500 mm2/min has resulted in stronger forces acting on the wire. These forces, together with the low rigidity of the wire, especially in the cutting of parts of high thickness, are responsible for wire deformation that has a direct influence on the accuracy of the part, mainly on wall-flatness and corners.
Paper can only be accepted if the discussion is better, based on missed works with high quality machines.
Author Response
Please see the document attached.

Reviewer 5 Report
In general, the work touches upon the topical issue of processing structural materials by electroerosive cutting. This method is widely demanded for obtaining parts in precision machine and instrument making. And its comprehensive study will expand technological capabilities and improve the quality of products.
Overall assessment of the work sections:
1. In the first part of the work (which for some reason is called “background” instead of the generally accepted “introduction”), a fairly detailed review of the literature in the field of research on the EDM process is provided. The formulation of what has been done in the work is given.
2. The second section provides a detailed description of the methodology of the work and describes the materials. In general, everything is fine here, but there is one significant remark.
3. Results. There is only one table with the measurement results.
4. Discussion. When discussing the results, their analysis is not used in conjunction with known literature data. In general, the section is more like a description of the results of mathematical processing of experimental data with a small analysis of empirical dependencies.
5. Conclusions.
Notes:
1. The annotation in the last sentence does not indicate what the influence of these parameters is aimed at. This proposal needs to be revised.
2. In the first part of the work there is no specifically formulated goal of the work. Why were these studies carried out? What are they aimed at? What technical result did the authors want to obtain?
3. The second section does not specify a method for measuring the dimensions of the gap. How and with what equipment was the control carried out?
4. I would recommend that the third and fourth sections be combined into one general section - "Results and Discussion".
5. Conclusions. Lines 410-424 have nothing to do with conclusions. This is just another introduction. This needs to be removed. The structure of the conclusions needs to be redone. Make brief conclusions showing (including quantitatively) the influence of the factors under consideration on the width of the cut.
Author Response
Please see the document attached

Round 2
Reviewer 1 Report
Following the changes made by the authors in the submitted manuscript entitled Shape errors of the surfaces obtained by WEDM, it can be published in the journal Micromachines.
I would like to thank the editor for allowing me to review this type of work and the authors for their research efforts.
Reviewer 2 Report
No
Reviewer 3 Report
The article has changed significantly in content and structure. In this form, the article can be accepted. Thanks to the authors for the work and correction of comments.
Reviewer 4 Report
Ok
Reviewer 5 Report
The authors took into account the main comments and made the necessary changes to the text of the article.
I believe that in its current form the article can be accepted for publication.